# Prevalence and correlates of pregnancy self-testing among pregnant women attending antenatal care in western Kenya

Nina Nganga[1], Julia Dettinger[1], John Kinuthia[1,2], Jared Baeten[1,3,4], Grace John-Stewart[1,3,4,5], Laurén Gómez[1], Mary Marwa[2], Ben Ochieng[6], Jillian Pintye[1], Kenneth Mugwanya[1], Melissa Mugambi[1]*

**1** Department of Global Health, University of Washington, Seattle, Washington, United States of America, **2** Department of Research and Programs, Kenyatta National Hospital, Nairobi, Kenya, **3** Department of Epidemiology, University of Washington, Seattle, Washington, United States of America, **4** Department of Medicine, University of Washington, Seattle, Washington, United States of America, **5** Department of Pediatrics, University of Washington, Seattle, Washington, United States of America, **6** University of Nairobi, Nairobi, Kenya

☯ These authors contributed equally to this work.

* mugambi@uw.edu

**Data Availability Statement:** All relevant data are available on Github: https://github.com/jcdettin/PrIMA-PregnancySelftesting.

## Abstract

In sub-Saharan Africa little is known about how often women use pregnancy self-tests or characteristics of these women despite evidence that pregnancy self-testing is associated with early antenatal care (ANC) initiation. Understanding the characteristics of women who use pregnancy self-tests can facilitate more targeted efforts to improve pregnancy testing experiences and entry into the ANC pathway. We conducted a cross-sectional survey among pregnant women enrolling in a pre-exposure prophylaxis (PrEP) implementation study to determine the prevalence and factors associated with pregnancy self-testing among women in western Kenya. Overall, in our study population, 17% of women obtained a pregnancy self-test from a pharmacy. Pregnancy test use was higher among employed women, women with secondary and college-level educated partners, and women who spent 30 minutes or less traveling to the maternal and child health (MCH) clinic. The most reported reasons for non-use of pregnancy self-tests included not thinking it was necessary, lack of knowledge, and money to pay for the test. Future research should focus on understanding the knowledge and attitudes of women toward pregnancy self-testing as well as developing community-based models to improve access to pregnancy testing and ANC.

## Introduction

The World Health Organization (WHO) recommends that pregnant women should initiate the first antenatal care (ANC) visit in the first trimester of pregnancy because early ANC access is central to identifying pregnancy complications and managing pre-existing conditions [1]. However, in western Kenya, less than 20% of pregnant women are estimated to present for ANC in the first trimester [2]. Barriers to early initiation of ANC due to uncertainty of

**Funding:** This work was supported by the research grant, NIH NIAID 1R01AI125498 (GJ-S and JB) and a Diversity Supplement award, NIH NIAID 3R01AI125498-03S1 (MM). Gilead Sciences provided support in the form of salaries for JMB. Pfizer provided support in the form of salaries for BH. The specific roles of these authors are articulated in the 'author contributions' section. The funders had no role in the study design, data collection, and analysis, decision to publish, or preparation of the manuscript.

**Competing interests:** JMB reports personal fees from Gilead Sciences, Janssen, and Merck, outside the submitted work. There are no patents, products in development, or marketed products associated with this research to declare. This does not alter our adherence to PLOS ONE policies on sharing data and materials.

pregnancy status during the first trimester can potentially be addressed by improving access to pregnancy testing [3, 4]. However, little is known about how often women use pregnancy self-tests or the characteristics of these women. Understanding the characteristics of women who use pregnancy self-tests can facilitate more targeted efforts to improve pregnancy testing experiences and entry into the ANC pathway. In this study, our primary objective was to determine the prevalence of pregnancy self-testing and associated factors among pregnant women attending maternal and child health (MCH) clinics in western Kenya. In a secondary objective, we evaluated whether pregnancy self-testing was associated with early ANC initiation among pregnant women.

## Methods

### Study design

From December 2018 to July 2019, we collected data as part of a baseline survey among pregnant women enrolling in the PrEP Implementation for Mothers in Antenatal Care (PrIMA) study. PrIMA is a cluster randomized trial (NCT03070600) that compares facility-based approaches for delivering oral pre-exposure prophylaxis (PrEP) in pregnancy. Briefly, between January 2018 and July 2019, study participants were recruited from women presenting for ANC in 20 public health facilities in Homabay and Siaya counties in western Kenya. In this region, approximately more than 95% of pregnant women receive antenatal care from a skilled provider [2]. Health facilities eligible for inclusion in the PrIMA study were located in counties with an HIV prevalence of at least 20%, had more than 350 HIV-negative clients receiving ANC annually, and provided postnatal care services, including infant immunizations. Women were eligible for enrollment if they were: (1) pregnant, (2) HIV negative (based on test results abstracted from the MCH card), (3) not currently using PrEP, (4) ≥ 15 years of age, (5) TB negative and (6) planned to receive postnatal and infant care at the enrollment facility. Participants were recruited for the study while waiting to access ANC services at any ANC visit and at any gestational age. The study team administered the enrollment questionnaire in English, Kiswahili, or the local language, Dholuo (see S1 Appendix). Participants answered questions on socio-demographics, medical and pregnancy history, and partner characteristics. Additional details on the PrIMA study protocol are described elsewhere [5].

### Study variables

We analyzed two dependent variables: "obtained pregnancy self-test from pharmacy" and "early ANC initiation." If a woman reported using a pregnancy test on her own to confirm her pregnancy and obtained the pregnancy test from the pharmacy, we categorized her as having obtained a pregnancy self-test from the pharmacy. We defined "early ANC initiation" as starting ANC during the first trimester of pregnancy (gestational age less than 13 weeks). We analyzed variables that we hypothesized from the literature would be associated with pregnancy self-test use and early ANC: socio-demographics (age, partner status, number of years in school, partner's education level, and history of prior pregnancy), accessibility (travel time to the health facility), affordability (women's employment status) and women's knowledge or attitudes (history of pregnancy or delivery complications) [3, 4, 6–9]. We analyzed "obtained pregnancy self-test from pharmacy" as an independent variable when evaluating factors associated with early ANC initiation [3, 4]. We categorized age into three groups: 15–19 years, 20–24 years, and > 25 years. We included three different age groups because adolescents and young women face unique challenges in pregnancy and are likely to have different health-seeking behaviors [10]. We defined and categorized partner status into two groups: having a partner (including a married partner) or not having a partner in the last three months at the time of

the survey. The number of years in school was categorized into three groups (8 years, 9–12 years, and >12 years), reflecting the number of years that individuals have typically spent in school at primary, secondary, and college levels in the region. We categorized the partner's education level into four groups: primary school and below (no formal schooling or having attended or completed primary school), secondary school (having attended or completed secondary school), and college (having attended or completed college). We defined history of prior pregnancy as a woman reporting that she had previously been pregnant. Women reported estimated travel time to the health facility in minutes which we categorized into two groups ($\leq 30$ minutes versus $> 30$ minutes) based on the median travel time. Women were categorized as employed if they reported currently having regular employment. Women had a history of prior pregnancy or delivery complications if they indicated that they had the following problems during her last pregnancy: late pregnancy bleeding, miscarriage, fetal miscarriage, high blood pressure, laceration, hemorrhage, obstructed labor, premature birth (<37 weeks), spontaneous fetal death (< 20 weeks) or stillbirth. All variables were self-reported.

## Statistical analysis

We conducted a descriptive analysis of the independent (age, partner status, history of prior pregnancy, employment status, and the number of years in school) and dependent variables (obtained pregnancy self-test from pharmacy and early ANC initiation). We examined the sources of pregnancy self-tests and the reasons for non-use of pregnancy self-tests among pregnant women. We conducted bivariate logistic regression analyses to examine the associations between the independent variables and each of the two dependent variables. In the multivariable logistic regression analysis, we excluded age and number of years in school because they were strongly correlated with a history of prior pregnancy and partner's education level, respectively (Cramer's $V > 0.30$, $p < 0.05$) [11]. To maintain the sample size, we excluded history of pregnancy or delivery complications because we could only determine this variable among women who had previously been pregnant. We included women who reported not having a partner as a separate category under the partner's education level. We used the Mantel-Haenszel test of homogeneity to identify potential modifiers of the relationship between obtaining a pregnancy self-test from a pharmacy and early ANC initiation. The relationship differed significantly by partner's education level; therefore, we stratified the analysis by this variable. Given that we conducted a secondary analysis of data from an existing cluster randomized trial, we adjusted for the 20 facility clusters using a robust method of standard errors. Overall, 93% of records had complete data. Missingness ranged from 0.6% to 4.6% across all variables. Statistical analyses were performed using R software (R-Studio Version 1.1.456) and STATA 16.1 (College Station, TX).

## Ethics

The Kenyatta National Hospital (P73/02/2017) and the University of Washington (STUDY00000438) institutional review boards approved this study. All participants provided written informed consent to participate in the study.

## Results

### Socio-demographic characteristics

From December 10th, 2018, to July 31st, 2019, 1128 women enrolled in the PrIMA study and completed the baseline survey. We excluded six records where the participants reported presenting for ANC earlier than the trimester in which they confirmed their pregnancy. Overall,

this analysis included 1122 pregnant women between the ages of 15–43 years with a median age of 24 years (interquartile range [IQR]: 21–28 years). At the time of the survey, the majority of women were married or had partners (92%), had previously been pregnant (78%), were employed (87%), and had completed up to 12 years of school (84%).

## Prevalence and correlates of pregnancy self-testing

Table 1 summarizes the characteristics of pregnant women who confirmed their pregnancy with or without obtaining a pregnancy self-test from a pharmacy. Seventeen percent of women reported obtaining a pregnancy self-test from a pharmacy to confirm their pregnancy. An additional 5% of women reported using a pregnancy self-test and obtained their kits from a public health facility (n = 36), a private health facility (n = 17), and other sources including

**Table 1. Characteristics of pregnant women who confirmed their pregnancy with or without obtaining a pregnancy self-test from a pharmacy.**

| Variables | Obtained a pregnancy self-test from the pharmacy N = 187 * | | Did not obtain a pregnancy self-test from the pharmacy N = 903 * | | |
|---|---|---|---|---|---|
| | N | (%) | N | (%) | p-value |
| **Maternal age** | | | | | |
| 15–19 | 18 | (10) | 131 | (15) | 0.193 |
| 20–24 | 76 | (42) | 352 | (41) | |
| ≥25 | 86 | (48) | 382 | (44) | |
| **Number of years in school** | | | | | |
| ≤ 8 | 43 | (23) | 397 | (44) | <0.001 |
| 9–12 | 79 | (43) | 388 | (43) | |
| > 12 | 63 | (34) | 108 | (12) | |
| **History of prior pregnancy** | | | | | |
| Yes | 134 | (72) | 716 | (79) | 0.02 |
| No | 53 | (28) | 186 | (21) | |
| **Partner status** | | | | | |
| Have a partner or married | 171 | (92) | 824 | (92) | 0.937 |
| No partner | 15 | (8) | 74 | (8) | |
| **Currently employed** | | | | | |
| Yes | 46 | (25) | 92 | (10) | <0.001 |
| No | 139 | (75) | 800 | (90) | |
| **Travel time to the health facility (minutes)** | | | | | |
| ≤30 | 140 | (76) | 524 | (59) | <0.001 |
| >30 | 45 | (24) | 369 | (41) | |
| **Partner's education level** | | | | | |
| Primary school and below | 20 | (12) | 310 | (38) | <0.001 |
| Secondary school | 71 | (42) | 341 | (41) | |
| College | 80 | (47) | 173 | (21) | |
| **History of pregnancy or delivery complications** | | | | | |
| Yes | 13 | (10) | 68 | (10) | 0.9 |
| No | 117 | (90) | 637 | (90) | |
| **Timing of presentation for ANC** | | | | | |
| 1st trimester | 80 | (43) | 292 | (32) | 0.009 |
| 2nd trimester | 98 | (52) | 528 | (59) | |
| 3rd trimester | 9 | (5) | 82 | (9) | |

*The number of respondents for each variable may vary due to missing responses.

community health workers, school, and home (n = 4). Of the 846 respondents who did not use a self-test: 85% confirmed their pregnancies at a public health facility, 8% at a private health facility, and 7% used other means. Among women who reported using other means to confirm their pregnancies: 43 stated that they did not confirm their pregnancies or knew they were pregnant based on physical signs of pregnancy, including missed periods, fatigue, nausea, and abdominal growth, eight went to traditional birth attendants, community health workers or a midwife and two went to an unspecified health facility. The most frequent reasons for non-use of pregnancy self-tests included: not thinking it was necessary (58%), lack of knowledge on self-tests (25%), and lack of money to pay for a self-test (11%).

Table 2 shows the univariate and multivariable logistic regression results for correlates of pregnancy self-test use. In the univariate analysis, women who obtained a pregnancy self-test from the pharmacy were more likely to be pregnant for the first time, employed, more educated (9–12 years or > 25 years vs. ≤ 8 years), spend 30 minutes or less traveling to the health facility, and have more educated partners (secondary school or college vs. primary school or below). In the multivariable analysis, women who obtained a pregnancy self-test from the pharmacy were more likely to be employed, spend 30 minutes or less traveling to the health facility, and have more educated partners (secondary school or college vs. primary school or

**Table 2. Univariate and multivariable logistic regression examining the variables associated with obtaining a pregnancy self-test from a pharmacy.**

| Variables | Univariate Model | | Multivariable Model | |
|---|---|---|---|---|
| | OR (95% CI) | p-value | OR (95% CI) | p-value |
| **Partner status** | | | | |
| Have a partner or married | 1.02 (0.46, 2.26) | 0.954 | - | - |
| No partner | Reference group | | - | - |
| **History of prior pregnancy** | | | | |
| Yes | Reference group | | Reference group | |
| No | 1.52 (1.13, 2.06) | 0.006 | 1.22 (0.87, 1.70) | 0.251 |
| **Age** | | | | |
| 15–19 | 0.61 (0.35, 1.07) | 0.087 | - | - |
| 20–24 | 0.96 (0.59, 1.55) | 0.87 | - | - |
| ≥25 | Reference group | | | |
| **Currently employed** | | | | |
| Yes | 2.88 (2.08, 3.97) | <0.0001 | 2.01 (1.31, 3.08) | 0.001 |
| No | Reference group | | Reference group | |
| **Number of years in school** | | | | |
| ≤ 8 years | Reference group | | | |
| 9–12 years | 1.87 (1.30, 2.71) | 0.001 | - | - |
| > 12 years | 5.39 (2.74, 10.57) | <0.0001 | - | - |
| **Travel time to health facility(minutes)** | | | | |
| ≤30 | 2.19 (1.40, 3.42) | 0.001 | 1.81 (1.19, 2.73) | 0.005 |
| >30 | Reference group | | Reference group | |
| **Partner's education level** | | | | |
| Primary school and below | Reference group | | Reference group | |
| Secondary school | 3.22 (1.66, 6.27) | 0.001 | 3.11 (1.63, 5.93) | 0.001 |
| College | 7.17 (3.92, 13.11) | <0.0001 | 6.11 (3.40, 10.96) | <0.0001 |
| No partner | - | - | 2.71 (1.22, 6.00) | 0.014 |
| **History of pregnancy or delivery complications** | | | | |
| Yes | 1.04 (0.67, 1.62) | 0.859 | - | - |
| No | Reference group | | - | - |

below). Obtaining a pregnancy self-test from the pharmacy was not associated with a history of a prior pregnancy.

## Prevalence and correlates of early antenatal care initiation

Approximately 65% of women confirmed their pregnancy in the first trimester. However, only 35% of the women initiated ANC early–in the first 13 weeks of pregnancy. Fifty-eight percent of the women started ANC during the second trimester, and 8% during the third trimester.

Table 3 shows the univariate logistic regression results for correlates of early ANC initiation. In the univariate analysis, women who initiated ANC early were more likely to have obtained a pregnancy self-test from a pharmacy, be pregnant for the first time, be employed, and have a more educated partner (college vs. primary school or below). Table 4 shows the multivariable logistic regression results stratified by partner education level. Early ANC initiation was more likely among women with college-educated partners if the women obtained a pregnancy self-test from a pharmacy (OR: 1.94, 95% CI: 1.12, 3.35). However, early initiation

**Table 3. Univariate logistic regression examining the variables associated with early ANC initiation.**

| Variables | OR (95% CI) | p-value |
|---|---|---|
| **Obtained pregnancy self-test from pharmacy** | | |
| Yes | 1.56 (1.12, 2.17) | 0.008 |
| No | Reference group | |
| **Partner status** | | |
| Have a partner or married | 0.67 (0.42, 1.09) | 0.105 |
| No partner | Reference group | |
| **History of prior pregnancy** | | |
| Yes | Reference group | |
| No | 1.55 (1.12, 2.13) | 0.008 |
| **Age** | | |
| 15–19 | 0.95 (0.65, 1.38) | 0.791 |
| 20–24 | 1.24 (0.98, 1.57) | 0.079 |
| ≥25 | Reference group | |
| **Currently employed** | | |
| Yes | 1.33 (1.01, 1.75) | 0.043 |
| No | Reference group | |
| **Number of years in school** | | |
| ≤ 8years | Reference group | |
| 9–12 years | 1.23 (0.97, 1.55) | 0.091 |
| > 12 years | 1.52 (0.96, 2.41) | 0.077 |
| **Travel time to health facility(minutes)** | | |
| ≤30 | 0.80 (0.55, 1.16) | 0.239 |
| >30 | Reference group | |
| **Partner's education level** | | |
| Primary school and below | Reference group | |
| Secondary school | 1.30 (1.00, 1.68) | 0.054 |
| College | 2.04 (1.56, 2.67) | <0.0001 |
| No partner | - | - |
| **History of pregnancy or delivery complications** | | |
| Yes | 1.35 (0.77, 2.37) | 0.300 |
| No | Reference group | |

**Table 4. Logistic regression examining the variables associated with early ANC initiation and stratified by partner's education level.**

| | Partner's education level | | | | | |
|---|---|---|---|---|---|---|
| Variables | Primary school and below (N = 327) | | Secondary school (N = 406) | | College (N = 247) | |
| | OR (95% CI) | p-value | OR (95% CI) | p-value | OR (95% CI) | p-value |
| **Obtained pregnancy self-test from pharmacy** | | | | | | |
| Yes | 0.25 (0.068, 0.92) | 0.036 | 0.92 (0.57, 1.50) | 0.749 | 1.94 (1.12, 3.35) | 0.018 |
| No | Reference group | | | | | |
| **History of prior pregnancy** | | | | | | |
| Yes | Reference group | | | | | |
| No | 2.36 (1.23, 4.51) | 0.009 | 2.04 (1.12, 3.71) | 0.019 | 0.89 (0.54, 1.48) | 0.654 |
| **Currently employed** | | | | | | |
| Yes | 2.03 (0.76, 5.44) | 0.157 | 1.11 (0.57, 2.16) | 0.752 | 0.95 (0.47, 1.89) | 0.879 |
| No | Reference group | | | | | |
| **Travel time to health facility(minutes)** | | | | | | |
| ≤30 | 1.20 (0.88, 1.63) | 0.251 | 0.95 (0.55, 1.64) | 0.858 | 1.19 (0.59, 2.39) | 0.621 |
| >30 | Reference group | | | | | |

among women with partners who had at least a primary level education was less likely if the women obtained a pregnancy self-test from a pharmacy (OR:0.25, 95% CI: 0.068, 0.92). Among women with partners who had at least a primary or secondary level education, early ANC initiation remained more likely if women were pregnant for the first time.

## Discussion

In this study, we investigated the prevalence and correlates of pregnancy self-testing among pregnant women attending MCH clinics in western Kenya. Overall, the prevalence of pregnancy self-testing in the study population was low, with 23% of women reporting having used a pregnancy self-test to confirm their pregnancy and 17% of women having obtained a pregnancy self-test from a pharmacy to confirm their pregnancy. We report a slightly lower prevalence of pregnancy self-test use than a 2006 South African study in which 27% of ANC clients obtained a pregnancy self-test from a private pharmacy [4]. Interestingly, most women who did not use a pregnancy self-test either did not think it was necessary or did not know that they could use one. Further studies are needed to understand women's knowledge and attitudes toward pregnancy self-testing.

In this study population, employment, education level, partner's education level, and travel time to health facility were the strongest correlates of pregnancy self-testing. Employed women may have greater autonomy in deciding how to spend financial resources than women who are not employed. Women who are more educated are more likely to have increased knowledge and familiarity with obtaining and using pregnancy tests [9]. Women with more educated partners might be likely to experience positive reinforcement of pregnancy testing behaviors and have the financial resources for pregnancy self-tests [12]. Finally, women who reported shorter travel times might be more likely to afford pregnancy self-tests or live in urban areas with better access to pregnancy self-tests [13]. Our findings are consistent with studies conducted in the United States that demonstrate increased rates of pregnancy self-test use among women of higher socioeconomic status [9].

In this study population, we found that 35% of women reported initiating ANC in the first trimester of pregnancy. This proportion is higher compared to findings reported for sub-Saharan Africa (24.9%) in a 2017 systematic review [14] and for the greater study region (<20%) in the 2014 Kenya Demographic and Health Survey [2]. In the univariate analysis, we found that

women who initiated ANC in the first trimester were more likely to obtain a pregnancy self-test from the pharmacy, have no history of prior pregnancy, be employed, and have a college-level educated partner. Interestingly, in the stratified analysis, we found that partner education level modified the association between obtaining a pregnancy self-test from the pharmacy and early ANC initiation. In cases where partners had a college-level education, women were twice as likely to initiate ANC early if they obtained a pregnancy self-test from a pharmacy. However, in cases where partners had a primary school education or below, women were less likely to start ANC early if they obtained a pregnancy self-test from a pharmacy. Prior studies show that a partner's education level is an essential determinant of women's health-seeking behavior [15, 16]. When partners are more educated, women are likely to live in households of higher socioeconomic status and able to forgo the opportunity costs of initiating ANC early [12].

To our knowledge this is one of the few studies to evaluate whether pregnancy testing is associated with early ANC initiation. One study in Ethiopia found that women who recognized their pregnancy using a urine test were more likely to initiate ANC early than women who used other means such as missed periods [17, 18]. A prior study in South Africa found that women who independently decided to obtain a urine pregnancy test from a private pharmacy initiated ANC 3.6 weeks earlier in their pregnancy [4]. However, neither of these studies adjusted for partner +*-education level; a recent systematic review from sub-Saharan Africa shows that studies did not evaluate the relationship between partner education and early ANC initiation [7]. Our study shows that pregnancy testing is an important prerequisite to initiating ANC early, particularly among women with highly educated partners. Strategies that increase knowledge and understanding of ANC and its benefits, bring care closer to the community, or provide financial and social support are important components to realizing the full benefits of pregnancy testing. For example, community-based models of care that combine the delivery of free pregnancy tests and home-based counseling could be a promising strategy for improving early ANC initiation [19, 20]. Exploring opportunities to engage community pharmacies, which are known to be frequently accessed [21] and sell pregnancy tests [22], could also expand the options available to women for early pregnancy recognition and referral to ANC [23].

Our study has some limitations. Although we were able to recruit over 1000 study participants from 20 clinics in western Kenya, the majority of the participants came from rural areas therefore some aspects of our findings may not generalize to other settings. Secondly, participants self-reported when they confirmed their pregnancy and when they first presented for antenatal care. It is therefore possible that some women may not accurately recall their pregnancy history leading to over- or underestimation of study outcomes. Thirdly, because our sample comprised women attending ANC as part of a PrEP implementation trial in public-sector MCH clinics, our findings may not be representative of all ANC attendees who use pregnancy self-tests in this setting. Finally, given that we conducted a cross-sectional analysis of enrollment data from an implementation trial, we were unable to assess additional independent variables such as knowledge and attitudes toward pregnancy self-testing and ANC [4].

## Conclusions

In conclusion, this study confirms that pregnancy testing plays a beneficial role in facilitating early ANC initiation. However, we found modest overall use of pregnancy self-tests among women attending ANC in western Kenya. Most women either did not see the utility of pregnancy self-testing or did not know about pregnancy self-tests. Further research is needed to understand women's attitudes, knowledge, and motivations toward pregnancy self-testing and how it informs decision-making around ANC attendance. To create value for women in the

ANC pathway, we need to explore strategies to adapt existing community-based models that include pregnancy testing, antenatal counseling, and timely referral for care.

## Supporting information

**S1 Appendix. Demographic and pregnancy history questionnaire.**
(PDF)

## Acknowledgments

We thank the PrEP Implementation for Mothers in Antenatal Care (PrIMA) study team and clients for their contributions to this study; we thank the Homa Bay and Siaya County Directors of Health for their support.

## Author Contributions

**Conceptualization:** Nina Nganga, Melissa Mugambi.

**Formal analysis:** Nina Nganga, Kenneth Mugwanya, Melissa Mugambi.

**Methodology:** Nina Nganga, Kenneth Mugwanya, Melissa Mugambi.

**Project administration:** Jillian Pintye.

**Supervision:** Jillian Pintye, Kenneth Mugwanya, Melissa Mugambi.

**Writing – original draft:** Nina Nganga, Melissa Mugambi.

**Writing – review & editing:** Nina Nganga, Julia Dettinger, John Kinuthia, Jared Baeten, Grace John-Stewart, Laurén Gómez, Mary Marwa, Ben Ochieng, Jillian Pintye, Kenneth Mugwanya, Melissa Mugambi.

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
