## [Decision Letter · Decision Letter 0]

1 Feb 2021

PONE-D-20-39327

Prevalence and correlates of pregnancy self-testing among pregnant women attending antenatal care in western Kenya

PLOS ONE

Dear Dr. Mugambi,

Thank you for submitting your manuscript to PLOS ONE. After careful consideration, we feel that it has merit but does not fully meet PLOS ONE’s publication criteria as it currently stands. Therefore, we invite you to submit a revised version of the manuscript that addresses the points raised during the review process.

There is potential for selection bias (also called collider bias or Berksonian bias) in this study, and this may have led to spurious associations. If the socio-demographic characteristics of women who attended the clinics and participated in the PrIMA trial are different from those who did not attend the clinics and participated in the trial, then assessing the socio-demographic variables in multivariable regression introduces Berksonian bias.  This is because the study is conditioned on attending the clinic and participating in the trial. The socio-demographic characteristics may influence clinic attendance and trial participation. Moreover, women  who self-test are more likely to attend ANC.  Thus, clinic attendance is a “collider” and given that the study is conditioned on this, the result is the bias I have mentioned above.

The authors have not provided sufficient background information to judge whether this bias indeed affected/or did not affect this study. What is the coverage of ANC in the study area? Women were eligible for recruitment during which ANC visit? (The first ANC visit or any ANC visit?).  Of the women attending ANC, what proportion participated in the trial? What proportion  of women in the study area attend ANC in the included health facilities?

The study methodology is scanty. The authors should refer to the STROBE checklist for reporting cross-sectional studies STROBE Statement: Available checklists (strobe-statement.org).

These are major concerns that threaten the validity of this study and it is important to address them adequately before a decision can be made on this manuscript.

We look forward to receiving your revised manuscript.

Kind regards,

Calistus Wilunda, DrPH

Academic Editor

PLOS ONE

3. Please include additional information regarding the survey or questionnaire used in the study and ensure that you have provided sufficient details that others could replicate the analyses. For instance, if you developed a questionnaire as part of this study and it is not under a copyright more restrictive than CC-BY, please include a copy, in both the original language and English, as Supporting Information.  If the original language is written in non-Latin characters, for example Amharic, Chinese, or Korean, please use a file format that ensures these characters are visible.

4. Please state whether you validated the questionnaire prior to testing on study participants. Please provide details regarding the validation group within the methods section.

Reviewers' comments:

Reviewer's Responses to Questions

**Comments to the Author**

1. Is the manuscript technically sound, and do the data support the conclusions?

Reviewer #1: Yes

Reviewer #2: Partly

2. Has the statistical analysis been performed appropriately and rigorously? 

Reviewer #1: Yes

Reviewer #2: No

3. Have the authors made all data underlying the findings in their manuscript fully available?

Reviewer #1: Yes

Reviewer #2: No

4. Is the manuscript presented in an intelligible fashion and written in standard English?

Reviewer #1: Yes

Reviewer #2: Yes

5. Review Comments to the Author

Reviewer #1: 1. Is the manuscript technically sound, and do the data support the conclusions?

The manuscript is technically sound as it addresses an area of public health interest-pregnancy self testing. However while the conclusion on main outcome is in tandem with data, the conclusion on correlates seems to be based on exploratory data(line 186/187) where the author explored reasons for not self testing for pregnancy instead of correlates assessed in regression

2.Has the statistical analysis been performed appropriately and rigorously?

Yes, appropriate technique applied ie logistic regression and confounders controlled for. To add more rigour to the analysis and show the reliability of the prevalence estimates reported, please add 95% CIs to these prevalence estimates

3. Have the authors made all data underlying the findings in their manuscript fully available?

Data available on request and contacts provided

4. Is the manuscript presented in an intelligible fashion and written in standard English?

Standard English has been used. The following revisions are necessary

-Line 10 is not clear

-To improve logical flow, results on outcomes(ie the prevalence estimates) should be removed from the section on sociodemographics and brought down to the respective sections where the correlates are being presented

Reviewer #2: 1. Line 41- variables that were hypothesized should be based on previous studies in similar context. References should be provided.

2. Statistical analysis- What was the criteria of including a variable in the multi-variable analysis after conducting the univariate ? Was multi-collinearity assessed. Provide results for the same.

3. Was the travel time self-reported ? discuss its limitations. Further travel time and the location of the health facility are likely be highly correlated and infer the same thing

4. Ethics- provide the study protocols review numbers also in this section

5. Summarise the missing responses in the data and reflect on how it might have affected the analysis

6. PLOS authors have the option to publish the peer review history of their article (what does this mean?). If published, this will include your full peer review and any attached files.

Reviewer #1: No

Reviewer #2: No

---

## [Author Response · Author response to Decision Letter 0]

17 Sep 2021

Response to Reviewers

Re: Prevalence and correlates of pregnancy self-testing among pregnant women attending antenatal care in western Kenya (. PONE-D-20-39327)

Editor comments

 There is potential for selection bias (also called collider bias or Bersanian bias) in this study, and this may have led to spurious associations. If the socio-demographic characteristics of women who attended the clinics and participated in the PrIMA trial are different from those who did not attend the clinics and participated in the trial, then assessing the socio-demographic variables in multivariable regression introduces Berksonian bias. This is because the study is conditioned on attending the clinic and participating in the trial. The socio-demographic characteristics may influence clinic attendance and trial participation. Moreover, women who self-test are more likely to attend ANC. Thus, clinic attendance is a “collider” and given that the study is conditioned on this, the result is the bias I have mentioned above.

Response: We appreciate the reviewer's comments that the characteristics of the women who attended ANC and enrolled in the PrIMA study could be different from the women who did not participate in ANC and were not enrolled. We have accounted for this potential bias in the discussion section. We include the following sentence: "Thirdly, because our sample comprised women attending ANC as part of a PrEP implementation trial in public-sector MCH clinics, our findings may not be representative of all ANC attendees who use pregnancy self-tests in this setting." [Please see revised manuscript page 16 lines 64 - 66]

PrIMA was intentionally designed to represent the underlying population of HIV-negative ANC clients in Homa Bay and Siaya counties by enrolling women attending any ANC visit at any gestational age. Additionally, prior studies show that only 2.6% of pregnant women in the region of PrIMA implementation did not present for ANC during pregnancy, indicating a high frequency of ANC attendance. Therefore, our study population is likely representative of the general population of pregnant women in this setting.

Additionally, we expanded on the process of facility and participant selection in our revised submissions. Our newly added sentence reads, “ Briefly, between January 2018 and July 2019, study participants were recruited from women presenting for ANC in 20 public health facilities in Homabay and Siaya counties in western Kenya. In this region, approximately more than 95% of pregnant women receive antenatal care from a skilled provider[2]. Health facilities eligible for inclusion in the PrIMA study were located in counties with an HIV prevalence of at least 20%, had more than 350 HIV-negative clients receiving ANC annually, and provided postnatal care services, including infant immunizations. Women were eligible for enrollment if they were: (1) pregnant, (2) HIV negative (based on test results abstracted from the MCH card), (3) not currently using PrEP, (4) ≥ 15 years of age, (5) TB negative and (6) planned to receive postnatal and infant care at the enrollment facility. Participants were recruited for the study while waiting to access ANC services any ANC visit and at any gestational age”. [Please see revised manuscript pages 3 – 4 lines 18 – 28].

 The authors have not provided sufficient background information to judge whether this bias indeed affected/or did not affect this study. What is the coverage of ANC in the study area? Women were eligible for recruitment during which ANC visit? (The first ANC visit or any ANC visit?). Of the women attending ANC, what proportion participated in the trial? What proportion of women in the study area attend ANC in the included health facilities?

Response: We appreciate the reviewers comment and have described the study background in more detail [please see revised manuscript pages 3-4 as well as the previous response]. As we described above, we have added this selection bias as a potential limitation in the discussion section. Findings from the 2014 KDHS reported that 96.6% of pregnant women in Nyanza region (where PrIMA was conducted) received antenatal care from a skilled provider (doctor, nurse or midwife). As described above, PrIMA was intentionally designed to represent the underlying population of HIV-negative ANC clients by enrolling women attending any ANC visit at any gestational age. Additionally, of the 9,915 HIV-negative women who received ANC at the health facilities, 8,391 women were screened for the study and thus we screened a high proportion of the overall HIV-negative ANC population. 

 The study methodology is scanty. The authors should refer to the STROBE checklist for reporting cross-sectional studies STROBE Statement: Available checklists (strobe-statement.org).

Response: We appreciate the reviewers comments and have carefully revised the methods section and modified the non-compliant parts according to the STROBE checklist for reporting cross-sectional studies. These changes are reflected in the revised manuscript on pages 3 – 6. 

Comments to the Author

1. Is the manuscript technically sound, and do the data support the conclusions?

Reviewer #1: Yes

Reviewer #2: Partly

2. Has the statistical analysis been performed appropriately and rigorously? 

Reviewer #1: Yes

Reviewer #2: No

3. Have the authors made all data underlying the findings in their manuscript fully available?

Reviewer #1: Yes

Reviewer #2: No

4. Is the manuscript presented in an intelligible fashion and written in standard English?

Reviewer #1: Yes

Reviewer #2: Yes

5. Review Comments to the Author

Reviewer #1: 

1. Is the manuscript technically sound, and do the data support the conclusions?

The manuscript is technically sound as it addresses an area of public health interest-pregnancy self-testing. However, while the conclusion on main outcome is in tandem with data, the conclusion on correlates seems to be based on exploratory data (line 186/187) where the author explored reasons for not self-testing for pregnancy instead of correlates assessed in regression

Response: We appreciate the reviewer considers our manuscript technically sound. In response to the reviewers comment about the conclusion we have updated this section to read: “ However, we found modest overall use of pregnancy self-tests among women attending ANC in western Kenya. Women who were employed, more educated, had more educated partners, and took a shorter time traveling to the health facility were more likely to use pregnancy self-tests. Most women either did not see the utility of pregnancy self-testing or did not know about pregnancy self-tests.” [Please see revised manuscript page 17, lines 72 – 76].

2. Has the statistical analysis been performed appropriately and rigorously?

Yes, appropriate technique applied i.e. logistic regression and confounders controlled for. To add more rigour to the analysis and show the reliability of the prevalence estimates reported, please add 95% CIs to these prevalence estimates

Response: We have reviewed the manuscript and included the 95% CIs to the odds ratios and key prevalence estimates. 

3. Have the authors made all data underlying the findings in their manuscript fully available?

Data available on request and contacts provided

Response: We have provided a limited and anonymized data set to a GitHub page, which can be accessed at the following URL (https://github.com/jcdettin/PrIMA-PregnancySelftesting).

4. Is the manuscript presented in an intelligible fashion and written in standard English?

Standard English has been used. The following revisions are necessary

-Line 10 is not clear

-To improve logical flow, results on outcomes (i.e. the prevalence estimates) should be removed from the section on sociodemographic and brought down to the respective sections where the correlates are being presented

Response: We agree with the reviewer and following changes have been made:

 Line 10, which was previously stated as: “Understanding the characteristics of women who use pregnancy self-tests is important to facilitate early access to ANC and to preventive interventions in pregnancy” has been restated as follows: “Understanding the characteristics of women who use pregnancy self-tests can facilitate more targeted efforts to improve pregnancy testing experiences and entry into the ANC pathway” [see lines 9 – 11 on page 3]

 The pregnancy related characteristics in the results section have been moved to their respective subheadings under “Prevalence and correlates of early antenatal care initiation” on page 11.

Reviewer #2:

 Line 41- variables that were hypothesized should be based on previous studies in similar context. References should be provided.

Response: We agree with the reviewer and have provided references on page 4 line 46.

2. Statistical analysis- What was the criteria of including a variable in the multi-variable analysis after conducting the univariate? Was multi-collinearity assessed. Provide results for the same.

Response: We appreciate the reviewers comments. Key inclusion criteria described in the methods section are as follows [Please see revised manuscript pages 4 – 5]:

 We analyzed variables that we hypothesized from the literature would be associated with pregnancy self-test use and early ANC. 

 Multicollinearity was assessed using Cramer’s V. We excluded age and number of years in school because they were strongly correlated with a history of prior pregnancy and partner’s education level, respectively (Cramer’s V > 0.30, p <0.05). 

 Age and history of prior pregnancy, Cramer’s V = 0.47

 Number of years in school and partner’s education level, Cramer’s V = 0.35

 To maintain the sample size, we excluded history of pregnancy or delivery complications because we could only determine this variable among women who had previously been pregnant. 

 We included women who reported not having a partner as a separate category under the partner’s education level. 

 We used the Mantel-Haenszel test of homogeneity to identify potential modifiers of the relationship between obtaining a pregnancy self-test from a pharmacy and early ANC initiation. The relationship differed significantly by partner’s education level; therefore, we stratified the analysis by this variable

 Given that we conducted a secondary analysis of data from an existing cluster randomized trial, we adjusted for the 20 facility clusters using a robust method of standard errors. 

3. Was the travel time self-reported? discuss its limitations. Further travel time and the location of the health facility are likely be highly correlated and infer the same thing

Response: Travel time to health facility was a self-reported variable and was captured by the question ‘How long does it take you to travel to clinic? (number of minutes)’. Travel time to health facility is dependent on the mode of transport to the health facility (walking, motor vehicle either personal or public or Boda-boda (bicycle or motorcycle taxis)). In the PrIMA study, the travel time question did not account for the various modes of transport.

We initially included location of health facility (urban/rural) as a variable to account for the 20 different facilities that were included in the study. We have now revised our approach and adjusted for the 20 health facility clusters using a robust methods of standard errors. 

4. Ethics- provide the study protocols review numbers also in this section

Response: The study was approved by the institutional review boards at the Kenyatta National Hospital (P73/02/2017) and the University of Washington (STUDY00000438). We have also included the review numbers in the manuscript.

5. Summarise the missing responses in the data and reflect on how it might have affected the analysis

Response: We have summarized the missing responses by study variable:

Study variable Number missing % missing

Early ANC initiation 12 1.1

Obtained pregnancy self-test at pharmacy 32 2.9

Maternal age 52 4.6

Partner status 11 1

Travel time to health facility 18 1.6

Currently employed 19 1.7

Number of years in school 20 1.8

Partner’s education level 10 0.9

History of prior pregnancy 29 2.6

Overall, 93% of records had complete data. Missingness ranged from 0.6% to 4.6% across all variables. Therefore it is highly unlikely that missing data had a substantial impact on our findings. 

6. PLOS authors have the option to publish the peer review history of their article (what does this mean?). If published, this will include your full peer review and any attached files.

Do you want your identity to be public for this peer review? For information about this choice, including consent withdrawal, please see our Privacy Policy.

Reviewer #1: No

Reviewer #2: No

Response: We have carefully revised our manuscript and modified the non-compliant parts according to the PLOS ONE style requirements.

Response: The study was approved by the institutional review boards at the Kenyatta National Hospital (P73/02/2017). and the University of Washington (STUDY00000438). All participants provided written informed consent to participate in the study. The study recruited pregnant women who were 15 years and older. Pregnant women (or women who have ever been pregnant) who are <18 years are considered ‘emancipated minors’ and are legally able to consent themselves for medical research.

3. Please include additional information regarding the survey or questionnaire used in the study and ensure that you have provided sufficient details that others could replicate the analyses. For instance, if you developed a questionnaire as part of this study and it is not under a copyright more restrictive than CC-BY, please include a copy, in both the original language and English, as Supporting Information. If the original language is written in non-Latin characters, for example Amharic, Chinese, or Korean, please use a file format that ensures these characters are visible.

Response: We have included a sample of the questionnaire used in the study as a supplementary file. 

4. Please state whether you validated the questionnaire prior to testing on study participants. Please provide details regarding the validation group within the methods section.

Response: We did not pilot test the questionnaire prior to using it for the study. However, most of the pregnancy history questions are asked as part of standard sociodemographic, pregnancy and postnatal care questions in the Demographic and Health Surveys (DHS) program, which is implemented by ICF International. A DHS model questionnaire can be accessed at: https://dhsprogram.com/pubs/pdf/DHSQ7/DHS7-Womans-QRE-EN-17Dec2018-DHSQ7.pdf.

Response: We have provided a minimal anonymized data set. We have addressed prompts a and b in our revised cover letter.

---

## [Decision Letter · Decision Letter 1]

1 Oct 2021

Prevalence and correlates of pregnancy self-testing among pregnant women attending antenatal care in western Kenya

PONE-D-20-39327R1

Dear Dr. Mugambi,

We’re pleased to inform you that your manuscript has been judged scientifically suitable for publication and will be formally accepted for publication once it meets all outstanding technical requirements.

Kind regards,

Calistus Wilunda, DrPH

Academic Editor

PLOS ONE

Additional Editor Comments (optional):

Reviewers' comments:

Reviewer's Responses to Questions

**Comments to the Author**

1. If the authors have adequately addressed your comments raised in a previous round of review and you feel that this manuscript is now acceptable for publication, you may indicate that here to bypass the “Comments to the Author” section, enter your conflict of interest statement in the “Confidential to Editor” section, and submit your "Accept" recommendation.

Reviewer #1: All comments have been addressed

Reviewer #2: All comments have been addressed

2. Is the manuscript technically sound, and do the data support the conclusions?

Reviewer #1: Yes

Reviewer #2: Yes

3. Has the statistical analysis been performed appropriately and rigorously? 

Reviewer #1: Yes

Reviewer #2: Yes

4. Have the authors made all data underlying the findings in their manuscript fully available?

Reviewer #1: Yes

Reviewer #2: Yes

5. Is the manuscript presented in an intelligible fashion and written in standard English?

Reviewer #1: Yes

Reviewer #2: Yes

6. Review Comments to the Author

Reviewer #1: (No Response)

Reviewer #2: (No Response)

7. PLOS authors have the option to publish the peer review history of their article (what does this mean?). If published, this will include your full peer review and any attached files.

Reviewer #1: No

Reviewer #2: No

---

## [Editor Report · Acceptance letter]

3 Nov 2021

PONE-D-20-39327R1 

Prevalence and correlates of pregnancy self-testing among pregnant women attending antenatal care in western Kenya 

Dear Dr. Mugambi:

I'm pleased to inform you that your manuscript has been deemed suitable for publication in PLOS ONE. Congratulations! Your manuscript is now with our production department. 

Kind regards, 

on behalf of

Dr. Calistus Wilunda 

Academic Editor

PLOS ONE